# Autism trends in a medium size coastal town of England

**Benjamin G. Fleet[1], Alicia Elliott[2], Margaret Orwin[2], Mark Spencer[3], Luigi Sedda[4]***

**1** Faculty of Health and Medicine, Medicine and Surgery MBChB Student at Lancaster Medical School, Lancaster University, Lancaster, United Kingdom, **2** NHS Midlands and Lancashire Commissioning Support Unit, Lancashire Business Park, Leyland, United Kingdom, **3** Healthier Fleetwood, Health & Wellbeing Centre, Fleetwood, United Kingdom, **4** Faculty of Health and Medicine, Lancaster Ecology and Epidemiology Group, Lancaster Medical School, Lancaster University, Lancaster, United Kingdom

* l.sedda@lancaster.ac.uk

## Abstract

Autism spectrum disorder (ASD) is a complex set of neurodevelopmental conditions which affects just under 1% of the global population. This study aims to investigate the trends in ASD diagnoses in a typical English deprived coastal community over the last two decades. ASD information for patients registered at Fleetwood GP practices were provided for the period between July 1952 to March 2022. The incidence and prevalence were calculated and Poisson regression modelling was employed to estimate the effects of age and sex on the number of ASD diagnoses over time. The study shows that there has been an upward trend in the number of ASD diagnoses over the past two decades. Model's results showed that sex differences in ASD diagnoses are less pronounced when accounting for time trends. The study findings show that Fleetwood has experienced a similar rise in ASD cases as the rest of the UK, most likely due to increased awareness that may explain the time effects over gender differences. However, due to the small sample size of the study, confirmation of the gender results and identification of the factors determining the temporal trends are needed in order to determine the gender effects in ASD diagnosis.

## Introduction

Autism spectrum disorder (ASD) is a complex set of neurodevelopmental conditions and is characterised by "early-onset difficulties in social communication and unusually restricted, repetitive behaviours and interests" [1]. Its aetiology is not fully understood although it is believed to be caused by a mix of genetic and environmental factors [2].

From a recent systematic review and meta-analysis, the global prevalence of ASD was estimated to be 0.72% [3].

The majority of those diagnosed with ASD are males with several studies reporting male to female ratios between 2:1 and 4:1 [1,4,5]. The exact reasons behind this difference are not yet understood, although part of this difference could be due to underdiagnosis in females for differences in presentation [6–14] which provides a challenge even to modern screening tests [15,16].

Within the UK, the prevalence is slightly higher than the rest of the world with most studies reporting a prevalence between 1 and 1.5% [4,17,18]. The UK male to female ratio in adults

only after ethical approval is granted by the requesting institution'.

**Funding:** This work was funded by the NHS, Wyre Council, Healthier Fleetwood, and Lancaster University ESPRC Impact Accelerator Account (reference: IAA: The use of innovative statistical methods aimed at improving the communities afflicted by COVID-19 patients). The funders had no role in study design, data collection and analysis, decision to publish, or preparation of the manuscript.

**Competing interests:** The authors have declared that no competing interests exist.

[19] is 3:1, a ratio within the global ranges. The UK has also experienced a steep increase in the incidence of ASD, by almost increasing 8-fold from 1998 to 2018 [20]. This trend could be a result of increased awareness towards ASD from both parents and clinicians [17,21].

Within the UK a large variation both in the overall ASD prevalence and in the ASD male to female ratio exists. This is in part due to inconsistencies between the tools used for diagnosis in different geographical areas [4], and geographical differences as shown in studies reporting increased urbanization and higher ASD diagnoses [22,23]. For the North West of England, the prevalence appears to be similar to other areas in England, albeit, with slightly higher prevalence in the more urbanized areas such as Liverpool and Manchester [4].

There is a lack of research around ASD in the North West of England and in coastal towns in general. For this reason, we have analysed the ASD data from a medium size coastal town in the North West of England. Fleetwood which itself is home to approximately 30,000 people, many of whom are experiencing poor health and are living in disadvantaged socioeconomic circumstances. Fleetwood are amongst some of the 10% most deprived areas and experiences lower educational attainment, greater levels of child and fuel poverty, higher prevalence of chronic health conditions, and lower life expectancy than nearby affluent areas [24]. A quarter of residents are living with a long-term illness (26% vs 18% England average) and mortality from most chronic conditions has been higher in Fleetwood than in the North West and England.

A risk for a higher than national ASD prevalence in Fleetwood may be inferred by the results of a recent global study that found children from socially disadvantaged families at an increased risk of developing ASD [3]. For towns like Fleetwood, it is mandatory to have a full understanding of the health challenges faced, which could shape local policies and promote further studies for other towns and cities in the North West and UK in general.

Here we present a study that aims to investigate the trends in ASD diagnoses in Fleetwood as typical English deprived coastal community. While findings are not generalisable at regional and national level, they can still provide key information in terms of at-risk groups and trends over the last two decades for public health interventions.

## Materials and methods

### Data

ASD information for patients registered at Fleetwood GP practices included information on patient age at diagnosis, year of diagnosis and sex. Data were provided for the period July 1952 to March 2022, although this analysis considered only the portion of data post-December 2001 due to the limited amount of cases being recorded before (5% of the whole dataset).

Total Fleetwood population per year and gender was acquired from the Middle Super Output Area (MSOA) population estimates published by the ONS between 2002 and 2020. The MSOA codes relevant for Fleetwood are: Wyre001, Wyre002, Wyre003 and Wyre005.

### Codes for autism

Two key resources exist for spectrum disorders: the Diagnostic and Statistical Manual of Mental Disorders (DSM-5) published in May 2013 [25] and the International Classification of Disease (ICD-11) published in May 2019 [26]. In this analysis we used the ICD classification.

As a result of changes in the diagnostic codes used to classify autism over time, we decided to aggregate the ASD codes into two classes, autism and Asperger's syndrome. Autism was used as an umbrella term and encompassed a variety of different diagnostic codes including autistic disorder, autistic spectrum disorder, autism spectrum disorder, infantile autism and childhood autism. There were fewer diagnostic codes for Asperger's syndrome, with the term including both Asperger and Asperger's syndrome.

### Ethics

Anonymised data was shared by each Fleetwood GP practice only considering patients that opted in the option to consent their data to be used anonymised for research purposes. All relevant ethical guidelines have been followed, and any necessary IRB and/or ethics committee approvals have been obtained. This protocol was approved by the Lancaster University Faculty of Health and Medicine Research Ethics committee (reference number FHMREC20121).

### Statistical analyses

Basic epidemiological statistics were calculated such as prevalence and incidence of ASD over the years. Odds ratios were employed to assess the current association between gender and each of the two ASD classes. These were calculated using both the prevalence and population data from the year 2020 as this was the most up to date data that was available.

Further statistical analysis was performed to assess how the ASD gender ratio changed over time. For this, temporally varying contingency tables were created and tested by chi-square test to compare gender (male/female) against ASD diagnosis (yes/no) for each year between 2002 and 2020.

Finally, a Poisson regression analysis assessed the changes in the number of diagnosis over time for both autism and Asperger's syndrome separately and then combined whilst controlling for gender and age.

Statistical analyses were performed using R statistical software version 4.2.0 (https://cran.r-project.org/).

## Results

Between 1st January 2002 and 31st December 2020 there were 33 diagnoses of 'Asperger's syndrome' and 163 diagnoses of 'Autism'. The gender split, male to female, of each group was 25/8 (24.2% female) and 126/37 (22.7% female) respectively. Overall, 23% of ASD diagnosis between 2002 and 2020 were female.

### Incidence and prevalence for autism and Asperger's syndrome

The prevalence of ASD rose from 0.07 to 7.03 cases per 1000 people (Fig 1). The overall incidence rose from 0.07 cases per 1000 people to 0.47 cases per 1000 (Fig 2). For males, the incidence rose from 0.15 to 0.66 cases per 1000 people whereas for females it rose from 0 to 0.28 cases per 1000 people (Fig 2). The prevalence of autism in Fleetwood rose from 0.04 cases per 1000 people to 5.85 cases per 1000 (S1 File). During this time the incidence rose from 0.04 cases per 1000 people to 0.43 cases per 1000 (S1 File). With regards to gender, the increase was from 0.07 to 0.59 cases per 1000 people for males and from 0 to 0.28 cases per 1000 people for females (S1 File). Within the same period the prevalence of Asperger's syndrome in Fleetwood increased from 0.04 to 1.18 cases per 1000 people (S1 File). No clear trends have been detected for Asperger's syndrome due to the limited amount of data available (S1 File).

The number of new autism and Asperger's diagnosis each year are shown in Fig 3.

### Odds ratios for the different classes by gender over time

In 2020, the odds of a confirmed autism diagnosis were over 3.5 times higher for males than females (OR 3.57, 95% CI: 2.47–5.15). A similar result was found in Asperger's syndrome diagnoses (OR 3.26, 95% CI: 1.47–7.22) (S1 File).

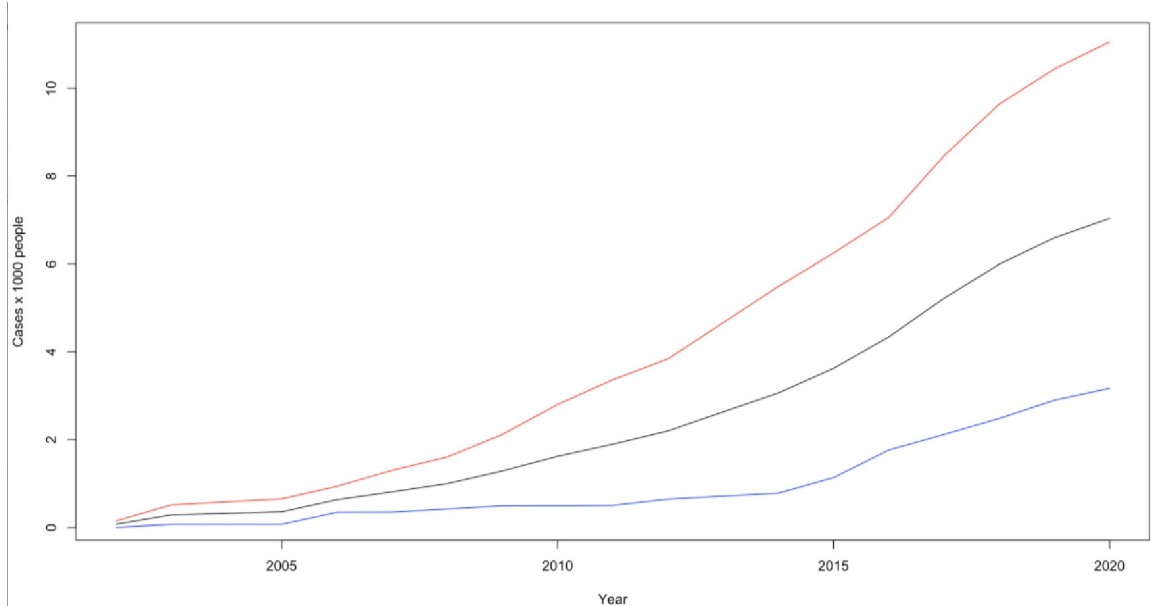

**Fig 1. Prevalence (per 1000 people) for ASD in Fleetwood from 2002 to 2020.** Lines legend: Black = combined male and female ASD prevalence, red = male ASD prevalence, blue = female ASD prevalence.

Over time, only in the last decade there were significant differences in the proportion of male and female diagnoses. In Fig 4 significant yearly differences in ASD gender proportion are shown (in red $p$-value lower than 0.05 using a chi-square test) (S1 File).

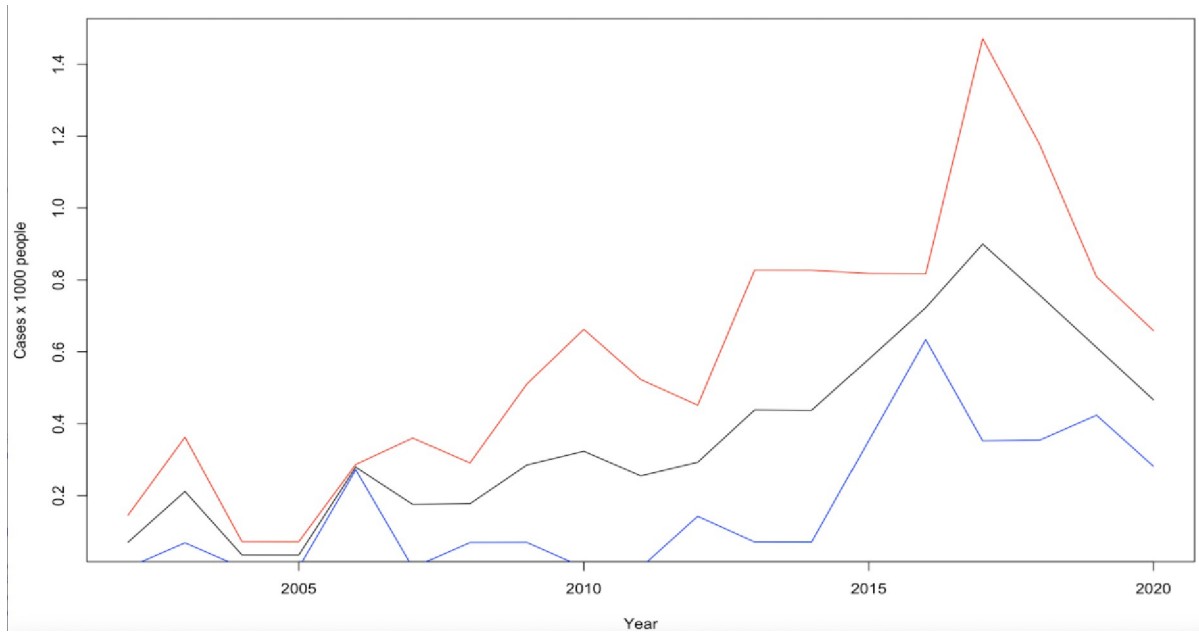

**Fig 2. Incidence (per 1000 people) for ASD in Fleetwood from 2002 to 2020.** Lines legend: Black = combined male and female ASD incidence, red = male ASD incidence, blue = female ASD incidence.

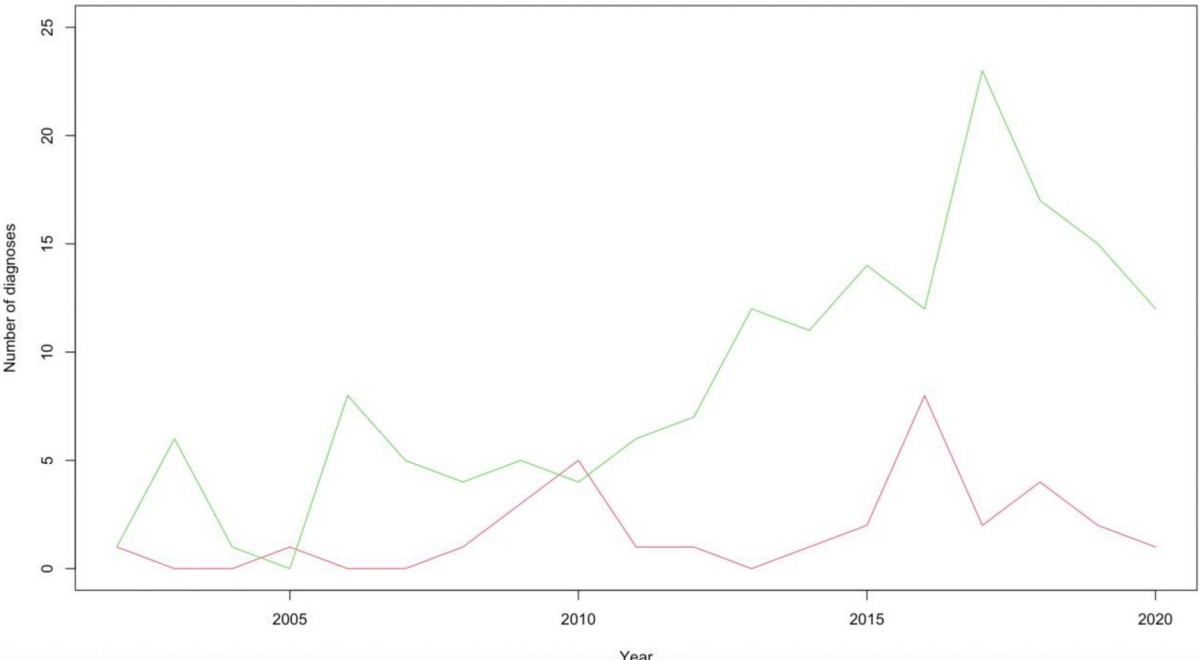

**Fig 3. New diagnoses for autism and Asperger's syndrome per year in Fleetwood from 2002 to 2020.** Lines legend: Green = autism, red = Asperger's syndrome.

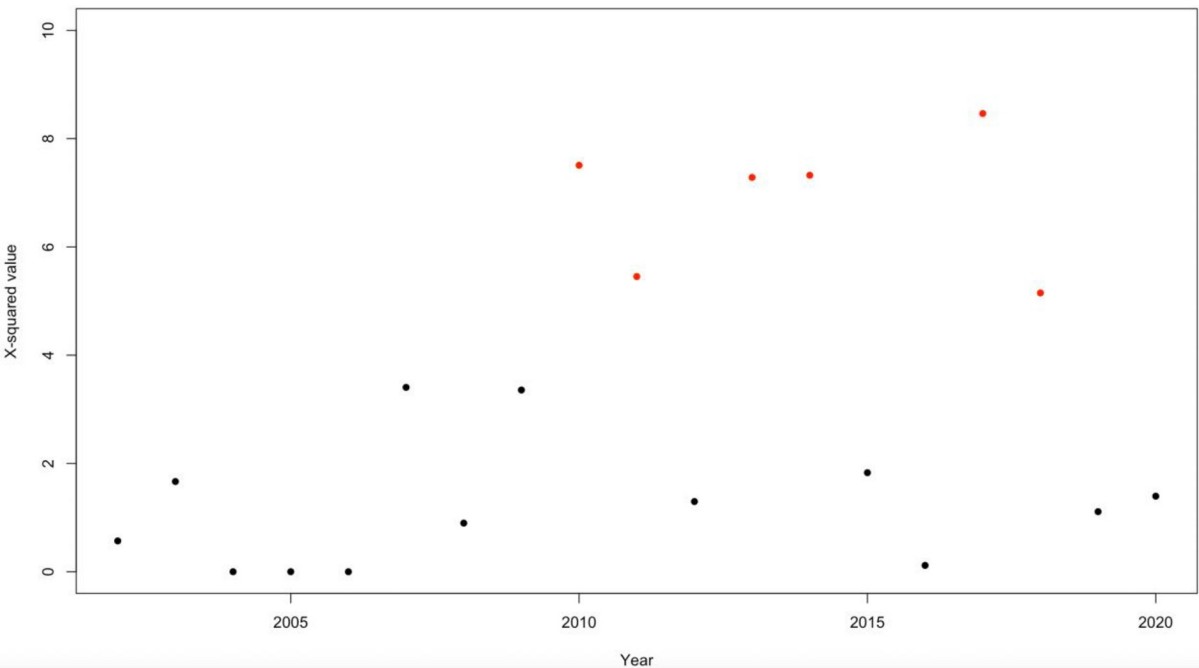

**Fig 4. Changes in gender proportion for all diagnoses from 2002 to 2020.** Points legend: Black = not significant, red = significant.

**Table 1. Coefficients for the logistic regression models for ASD diagnoses.**

| Variable | Coefficient | Standard Error | *p-value* |
|---|---|---|---|
| Model (1) | | | |
| Intercept | 1.125901e-95 | 29.10362 | 5.82e-14 |
| Year | 1.116044 | 0.01445 | 3.01e-14 |
| Model (2) | | | |
| Intercept | 1.778253e-71 | 38.76625 | 2.64e-05 |
| Year | 1.085445 | 0.01929 | 2.13e-05 |
| Gender | 1.077744 | 0.03453 | 0.0301 |

## Poisson regression analyses

Poisson regression results suggest that when time is taken as predictor, the effect of gender in the number of ASD diagnoses become slightly significant (*p-value* close to 0.05) (Table 1). This finding is more pronounced in the autism data than in the Asperger's syndrome diagnosis (S1 File).

## Discussion

Our study shows that there has been a clear upward trend in the number of ASD diagnoses over the past two decades in Fleetwood. During this time the prevalence has increased more than 20-fold and the incidence 7-fold. In 2020 the odds of having an ASD diagnosis were between 3 and 3.6 times higher for males than females. This finding is consistent with what has previously been reported in the literature [1,4,5]. The lack of diagnosis in females may be due to the difficulty to ascertain their symptoms [6,10,11]. For Asperger's syndrome, bearing the low numbers, we found a lower male to female ratio than what has previously been reported in the literature [27,28]. In additions, we found that only in the last decade significant differences in the number of males and females being diagnosed with ASD were present. It is important to highlight that the lack of significant differences in the preceding decade may have also been due to the low number of diagnoses, which increases the uncertainty in the statistical test. As reported elsewhere [20,21], it is difficult to ascertain the cause of these trends but it is likely that widespread public health campaigns and increased awareness may have contributed to them. Given the significance of the variable 'time' in our regression analyses, we suggest that the latter is to be considered as a major driver in the trends [29] making differences between genders less significant.

Even with this large increase in the number of cases, the overall prevalence of ASD in Fleetwood still remains below current global and UK prevalence estimates. However, it cannot be excluded that limitations in the provision of specialist support, or the deprivation itself, may have led to fewer diagnoses being made [4].

These results need to be taken with the caution of the study limitations. Firstly, the small sample size (a population size of around 30,000 people) and the specific conditions of Fleetwood (in terms of deprivation and other health and socio-economic measures) do not allow generalisation of the results–i.e. to regional or national levels. In addition, having considered only two classes representing a complex plethora of disorders could have reduced the capacity to discriminate between the true drivers of trends and gender spectrum ratios. Most importantly, due to identifiability restrictions it was not possible to associated other demographic and environmental factors that may be related to ASD diagnoses (genetic, environmental etc. . .) [30–34]. Nevertheless, the relationship between time, diagnoses and gender are robust and we expect that including more variables will only dilute the effect of gender in our models.

Finally, it is important also to recognise that when using diagnostic codes to identify patients, there is always a risk that the coding is inconsistent between users, or, that there have been errors in the medical records.

In conclusion, and on the light of recent national interest in coastal communities [35,36], we hope that these results can stimulate more research toward ASD especially in deciphering what is causing the ASD diagnosis gender ratio, since the risk is that especially female incidence is just the tip of the iceberg of a large hidden undiagnosed population. This work contributes to the recent debate around health and wellbeing in coastal communities which are now a priority for government interventions as stated in the UK Government 2020 analysis of Coastal Town, the 2021 report on 'Analysis of Health and Wellbeing in English Coastal Communities' and the 2022 Westminster Hall debate report on the 'Future of coastal communities'. Identifying prevalence, risk factors, risk groups, patterns and trends of ASD will allow effective quantification of resources and interventions aimed at improving the health and socio-economic status of the people affected by these disorders.

## Supporting information

**S1 File.** S1 Fig Prevalence (per 1000 people) for autism in Fleetwood from 2002 to 2020; S2 Fig Incidence (per 1000 people) for autism in Fleetwood from 2002 to 2020; S3 Fig Prevalence (per 1000 people) for Asperger's syndrome in Fleetwood from 2002 to 2020; S4 Fig Incidence (per 1000 people) for Asperger's syndrome in Fleetwood from 2002 to 2020; S1 Table Contingency table for autism diagnoses up to 2020; S2 Table Contingency table for Asperger's syndrome diagnoses up to 2020; S3 Table Annual chi-squared tests for gender between 2002–2020; S4 Table Coefficients summary for the logistic regression models for autism diagnoses; S5 Table Coefficients summary for the logistic regression models for Asperger's syndrome diagnoses.
(DOCX)

## Acknowledgments

We thank all the members of Healthier Fleetwood, Future Fleetwood, and the Fleetwood public for their support.

## Author Contributions

**Conceptualization:** Mark Spencer, Luigi Sedda.

**Data curation:** Benjamin G. Fleet, Alicia Elliott.

**Formal analysis:** Benjamin G. Fleet, Luigi Sedda.

**Investigation:** Benjamin G. Fleet, Luigi Sedda.

**Methodology:** Luigi Sedda.

**Project administration:** Luigi Sedda.

**Resources:** Alicia Elliott, Margaret Orwin, Luigi Sedda.

**Software:** Luigi Sedda.

**Supervision:** Margaret Orwin, Luigi Sedda.

**Validation:** Alicia Elliott, Mark Spencer.

**Visualization:** Benjamin G. Fleet.

**Writing – original draft:** Benjamin G. Fleet.

**Writing – review & editing:** Benjamin G. Fleet, Alicia Elliott, Margaret Orwin, Mark Spencer, Luigi Sedda.

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
