## [Decision Letter · Decision Letter 0]

3 Apr 2023

PONE-D-23-00169Autism trends in a medium size coastal town of EnglandPLOS ONE

Dear Dr. Sedda,

Thank you for submitting your manuscript to PLOS ONE. After careful consideration, we feel that it has merit but does not fully meet PLOS ONE’s publication criteria as it currently stands. Therefore, we invite you to submit a revised version of the manuscript that addresses the points raised during the review process.

We look forward to receiving your revised manuscript.

Kind regards,

Sebastien Kenmoe

Academic Editor

PLOS ONE

Journal Requirements:

2. Please ensure that you have specified (1) whether consent was informed and (2) what type you obtained (for instance, written or verbal, and if verbal, how it was documented and witnessed). If your study included minors, state whether you obtained consent from parents or guardians. If the need for consent was waived by the ethics committee, please include this information.

Reviewers' comments:

Reviewer's Responses to Questions

**Comments to the Author**

1. Is the manuscript technically sound, and do the data support the conclusions?

Reviewer #1: Yes

Reviewer #2: No

2. Has the statistical analysis been performed appropriately and rigorously? 

Reviewer #1: Yes

Reviewer #2: Yes

3. Have the authors made all data underlying the findings in their manuscript fully available?

Reviewer #1: Yes

Reviewer #2: Yes

4. Is the manuscript presented in an intelligible fashion and written in standard English?

Reviewer #1: Yes

Reviewer #2: Yes

5. Review Comments to the Author

Reviewer #1: The paper presents clear but minimal data about the prevalence and incidence of autism diagnoses from primary care health records from an economically deprived region in England. The major findings are that the prevalence and incidence are lower than expected from other reports and the rates increased in the last 10 years. However, the paper is limited by little information about the families or demographics of the sample other than the gender of the children. The numbers are also quite small so claims of a less pronounced sex difference could be due to the very small numbers of patients, particularly females. The paper is well-written and clear; the limitations are primarily the lack of other information.

Reviewer #2: The article provides a valuable contribution to the understanding of autism prevalence in Fleetwood, UK. However, there are several areas where the authors could improve their work to enhance its rigor and potential impact.

1) Authors should number the lines of their manuscript to facilitate the review process.

2) Introduction: To better support the global prevalence of autism, authors may cite the recent article with DOI 10.3389/fpsyt.2023.1071181. Authors should provide clear study objectives.

3) Methods: Authors should explain to what extent the population of Fleetwood represents the general population of England and whether the results of the study can be generalized to other regions.

4) Results: As the study only examines trends in one city, it is unclear whether the results are generalizable to other populations or geographic locations. The study relies on diagnostic codes recorded in medical records, which may be prone to errors or inconsistencies.

6. PLOS authors have the option to publish the peer review history of their article (what does this mean?). If published, this will include your full peer review and any attached files.

Reviewer #1: No

Reviewer #2: No

---

## [Author Response · Author response to Decision Letter 0]

13 May 2023

Full response provided in the reply to reviewers document.

---

## [Decision Letter · Decision Letter 1]

13 Jun 2023

Autism trends in a medium size coastal town of England

PONE-D-23-00169R1

Dear Dr. Sedda,

We’re pleased to inform you that your manuscript has been judged scientifically suitable for publication and will be formally accepted for publication once it meets all outstanding technical requirements.

Kind regards,

Sebastien Kenmoe

Academic Editor

PLOS ONE

Additional Editor Comments (optional):

Reviewers' comments:

Reviewer's Responses to Questions

**Comments to the Author**

1. If the authors have adequately addressed your comments raised in a previous round of review and you feel that this manuscript is now acceptable for publication, you may indicate that here to bypass the “Comments to the Author” section, enter your conflict of interest statement in the “Confidential to Editor” section, and submit your "Accept" recommendation.

Reviewer #2: All comments have been addressed

2. Is the manuscript technically sound, and do the data support the conclusions?

Reviewer #2: Yes

3. Has the statistical analysis been performed appropriately and rigorously? 

Reviewer #2: Yes

4. Have the authors made all data underlying the findings in their manuscript fully available?

Reviewer #2: Yes

5. Is the manuscript presented in an intelligible fashion and written in standard English?

Reviewer #2: Yes

6. Review Comments to the Author

Reviewer #2: (No Response)

7. PLOS authors have the option to publish the peer review history of their article (what does this mean?). If published, this will include your full peer review and any attached files.

Reviewer #2: No

---

## [Editor Report · Acceptance letter]

21 Jun 2023

PONE-D-23-00169R1 

Autism Trends in a Medium Size Coastal Town of England 

Dear Dr. Sedda:

I'm pleased to inform you that your manuscript has been deemed suitable for publication in PLOS ONE. Congratulations! Your manuscript is now with our production department. 

Kind regards, 

on behalf of

Dr. Sebastien Kenmoe 

Academic Editor

PLOS ONE